# AE-OT: A new Generative Model based on extended semi-discrete optimal transport

Dongsheng An[1], Yang Guo[1], Na Lei[✉2], Zhongxuan Luo[2], Shing-Tung Yau[3], and Xianfeng Gu[1]

[1]Department of Computer Science, Stony Brook University
[2]DUT-RU ISE, Dalian University of Technology
[3]Department of Mathematics, Harvard University
*{doan, yangguo}@cs.stonybrook.edu , {nalei, zxluo}@dlut.edu.cn , yau@math.harvard.edu,
gu@cs.stonybrook.edu*

## Abstract

Current generative models like generative adversarial networks (GANs) and variational autoencoders (VAEs) have attracted huge attention due to its capability to generate visual realistic images. However, most of the existing models suffer from the mode collapse or mode mixture problems. In this work, we give a theoretic explanation of the both problems by Figalli's regularity theory of optimal transportation maps. Basically, the generator compute the transportation maps between the white noise distributions and the data distributions, which are in general discontinuous. However, deep neural networks (DNNs) can only represent continuous maps. This intrinsic conflict induces mode collapse and mode mixture. In order to tackle the both problems, we explicitly separate the manifold embedding and the optimal transportation; the first part is carried out using an autoencoder (AE) to map the images onto the latent space; the second part is accomplished using a GPU-based convex optimization to find the discontinuous transportation maps. Composing the extended optimal transport (OT) map and the decoder, we can finally generate new images from the white noise. This AE-OT model avoids representing discontinuous maps by DNNs, therefore effectively prevents mode collapse and mode mixture.

## 1 Introduction

Generative adversarial networks (GANs) (Goodfellow et al. (2014)) and variational autoencoders (VAEs) (Kingma & Welling (2013)) emerge as the dominant approaches for unconditional image generation. When trained on appropriate datasets, they are able to produce realistic and visual appealing samples. GAN methods train an unconditional generator that regresses real images from random noise and a discriminator that measures the difference between generated samples and real images. Despite GANs' advantages, they have critical drawbacks. 1) Training of GANs are tricky and sensitive to hyperparameters. 2) GANs suffer from mode collapse, in which the generator only learns to generate few modes of data distribution while missing others, although samples from the missing modes occur throughout the training data (see e.g. Goodfellow (2016)). While for the VAEs, the encoder is used to map the data distribution to a Gaussian latent distribution, which is then mapped back to the data distribution by the decoder. While standard VAEs tend to capture all modes, they often generate ambiguous images on multimodal real data distributions. We propose that these phenomena relates deeply with the singularities of distribution transport maps.

**Manifold Distribution Hypothesis** In deep learning, the manifold distribution hypothesis is well accepted, which assumes the distribution of a specific class of natural data is concentrated on a low dimensional manifold embedded in the high dimensional data space Tenenbaum et al. (2000). Therefore, GANs and VAEs implicitly aim to accomplish two major tasks: 1) manifold embedding: to find the encoding/decoding maps between the data manifold embedded in the image space and the latent space; 2) probability distribution transport: to transport a given white noise distribution to the data distribution, either in the latent or in the image space.

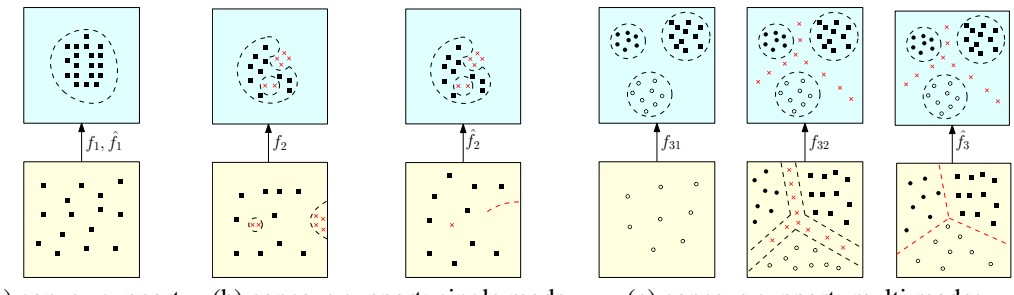

(a) convex support     (b) concave support: single mode     (c) concave support: multi modes

Figure 1: Mode collapse/mixture caused by the discontinuity of the transport map. Top row shows real data distributions, the bottom row gives the noise distributions. On top, each cluster represents a mode, the spurious generated samples are red crosses (mode mixture); at the bottom, red dotted lines are the singularity set, red crosses are mapped to be spurious samples by DNNs. (a) If the support of the target distribution is convex, DNN ($f_1$) is able to approximate the transport map $\hat{f}_1$ well. When the support of the target distributions are concave, there are two situations: (b) single mode and (c) multi modes. In (b), DNN, represented by $f_2$, cannot approximate the transport map $\hat{f}_2$ well and generates some spurious samples. $\hat{f}_3$ gives the transport map of multi-mode, when approximating it with continuous DNNs, either mode collapse $f_{31}$ or mode mixture $f_{32}$ will happen.

**Distribution Transformation** The generator of GAN model and the decoder of VAE model are trained to compute a transport map that transforms a known continuous distribution (e.g. Gaussian white noise) to the real data distribution. Namely, the transport map pushes forward the given noise distribution to a generated distribution to approximate the real data distribution, the similarity between the two distributions determines the generalization ability of the generator Ben-David et al. (2010).

**Discontinuity and Mode Collapse/Mixture** It is a common practice among GAN/VAE models that the generators/decoders are expressed by deep neural networks, which can only represent continuous mappings. Unfortunately, as pointed out by works Nagarajan & Kolter (2017); Khayatkhoei et al. (2018); Xiao et al. (2018), the transport maps may be discontinuous when there are multiple modes in the data distribution. This intrinsic conflict can cause mode collapse or mode mixture. The later means that the generated samples are mixtures of multiple modes and look spurious or ambiguous. [1]

Furthermore, even when the real data distribution has a *single mode*, ambiguous data (e.g. a human face image with mismatched eye colors) can still present. This can be explained by Brenier's polar factorization theorem Brenier (1991b; 1987; 1991a) and Figalli's regularity theorem Figalli (2010); Chen & Figalli (2017) (Thm. 5 in Appendix B), which asserts that if the support of the target distribution is not convex, then there will be *singularity sets* on the support of the source distribution, such that the transport map is discontinuous on these sets. This shows the intrinsic difficulties of conventional GANs/VAEs cannot be eliminated, as shown in Fig. 1.

**Conquering Mode Collapse/Mixture** However, according to Brenier (1987; 1991a) theorem, the optimal transport map can be represented as the gradient map of the Brenier potential. At the regular points, the Brenier potential is differentiable, its gradient map (the transport map) is continuous; at the singularities, the Brenier potential is continuous but not differentiable, and its gradient map is discontinuous. Conventional GANs and VAEs model the gradient map directly and encounter the trouble of discontinuity. In contrast, we propose to model the globally *continuous Brenier potential* to avoid mode collapse/mixture.

More specifically, our proposed AE-OT model separates the manifold embedding step and the probability distribution transformation step, the former is carried out by an autoencoder (AE), the latter is accomplished by a convex optimization framework (OT). The OT step computes the Brenier potential explicitly and is able to locate the singularity set (the discontinuous points of the gradient map) based on Figalli's theory. Our experimental results demonstrate that the proposed method can not only cover all of the modes, but also avoid generating spurious samples (mode mixture).

**Contributions** (i) From theoretical aspect, this work gives a thorough explanation of mode collapse and mode mixture by the regularity theory of optimal transportation developed by Figalli (2018

---

[1] For example, a generator generates obscure digits mixing 0 and 8 but neither 0 nor 8 on the MNIST dataset.

Fields medalist) and the reasons why standard GANs/VAEs cannot solve this problem perfectly. (ii) From practical aspect, this work proposes a novel model called AE-OT, which first encodes the data manifold into the latent space, then compute the Brenier potential to represent the optimal transportation map in the latent space. The Figalli's singularity set can be located efficiently and avoided when generating new samples. In this way, our model eliminates mode collapse and mode mixture successfully. (iii) The algorithm for finding the Brenier potential and the optimal transportation map can be accelerated with GPU based convex optimization algorithm. The method converges to the unique global optimum with bounded error estimate. (iv) Our experiment results demonstrate the efficiency and efficacy of the proposed method.

## 2 RELATED WORK

**Optimal Transport** Optimal transport (OT) has been successfully applied in the areas like economy, optics and machine learning, as surveyed in Solomon (2018) and Peyré & Cuturi (2018). In Gu et al. (2016) , the intrinsic connection between Brenier theory in OT and Alexandroff theory in convex geometry was established, and applied for deep learning in Lei et al. (2017) by an convex optimization. Figalli and the collaborators Figalli (2010); Chen & Figalli (2017) proposed that when the support of the data distribution is non-convex, the transport map will be discontinuous.

**Generative models** Generative model is one of the main tasks in the machine learning field. One of the most used image generation methods is Variational Autoencoders (VAEs), where the decoders approximate real data distributions from a Gaussian distribution in a variational approach (e.g Kingma & Welling (2013) and Rezende et al. (2014)). Later, Adversarial Autoencoders (AAEs) Makhzani et al. (2015) and Wasserstein Autoencoders (WAEs) Tolstikhin et al. (2018) are proposed following the similar scheme. Although VAEs are relatively simple to train, images they generate look blurry. Generative Adversarial Networks (GANs) Goodfellow et al. (2014) proposed by Goodfellow et.al can produce better quality images. While being a powerful tool in generating realistically looking samples, GANs suffer from the mode collapse problem. To solve this problem, a huge number of methods, including changing the loss function (e.g. Wasserstein GAN Arjovsky et al. (2017)), regularizing the discriminators to be Lipschitz (clipping Arjovsky et al. (2017), gradient regularization Gulrajani et al. (2017), Mescheder et al. (2018) or spectral normalization Miyato et al. (2018)), were proposed.

Besides, various non-adversarial methods has also been proposed recently. GLO Bojanowski et al. (2017) adopts the "encoder-less autoencoder" method to generate new images with a non-adversarial loss function. IMLE Li & Malik (2018) proposed an ICP related generative model training approach. Later GLANN Hoshen & Malik (2019) combines the advantages of GLO and IMLE, an embedding from image space to latent space was first found using GLO and then a transformation between an arbitrary distribution and latent code was computed using IMLE.

**Mitigating Mode Collapsing** Recently, Nagarajan & Kolter (2017); Khayatkhoei et al. (2018); Xiao et al. (2018) also realize the training difficulties of GANs come from the approximation of discontinuous functions with continuous DNNs. By the gradient-based regularization, GDGAN Nagarajan & Kolter (2017) do relieve the mode collapse phenomenon of GANs, but mode mixture still exists. Khayatkhoei et al. (2018) proposes to use multiple GANs to overcome the mode collapse. Xiao et al. (2018) proposed to embed the images into a latent space according to Bourgain's theorem, and train the generator by sampling a Gaussian mixture distribution in the latent space instead of a unimodal Gaussian. The recently introduced normalized diversification by Liu et al. (2018) can also help overcome mode collapse successfully. However, all of them cannot solve the mode mixture well.

All these works Nagarajan & Kolter (2017); Khayatkhoei et al. (2018); Xiao et al. (2018) explain that if the target data distribution has multiple modes, the transport map is discontinuous, but DNNs can only represent continuous mappings, the intrinsic conflict causes mode collapse.

## 3 COMPUTATIONAL ALGORITHMS

**Overview of AE-OT Model** Our AE-OT model is summarized in Fig. 2, it has two major components: i) (AE) An autoencoder is trained to encode ($f_\theta$) the data manifold from the image space $\mathcal{X}$ to the latent space $\mathcal{Z}$, and map the data distribution to the latent code distribution; then the decoder $g_\xi$ decodes the latent code back to the data manifold. ii) (OT) This module computes the optimal transportation map $T$ from the noise distribution to the latent code distribution. First, the Brenier potential is found by a convex optimization process according to Gu et al. (2016), whose gradient is the semi-discrete optimal transport map, where the target is the discrete set of latent codes of training

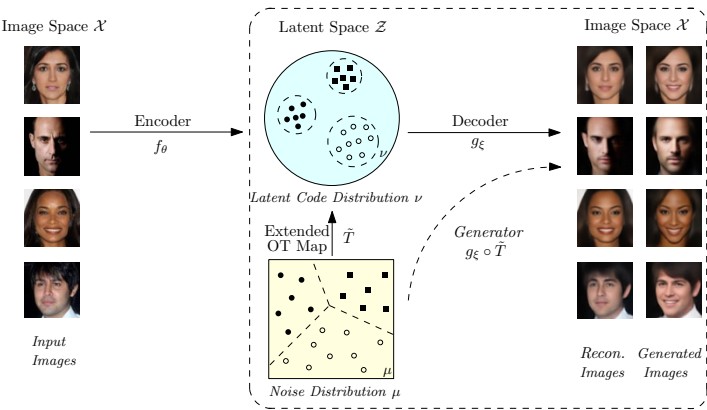

Figure 2: AE-OT model. AE: $f_\theta$ and $g_\xi$ represent the encoding and decoding maps respectively, where $\theta$ and $\xi$ are their corresponding network parameters. In the latent space $\mathcal{Z}$, the latent codes are clustered into three different modes, represented as marks with different shapes (i.e. disks, squares and circles). OT: The singular set between different modes is plotted with dashed lines. Finally, the generator of our model, which generates realistic images from random noise samples, is the composition of the extended OT map $\tilde{T}$ and the decoding map $g_\xi$.

samples; then the transport map is piece-wise linearly extended to a global continuous map $\tilde{T}$, where the image domain becomes a simplicial complex obtained by triangulating the above latent codes. Finally, the singularity set in the source domain is located and avoided when generating new samples. As a result, given a random noise $x$, we can get the generated image by $g_\xi \circ \tilde{T}(x)$.

**Semi-Discrete OT Map** Suppose the source measure $\mu$ (Gaussian or uniform distribution) is absolutely continuous defined on a convex domain $\Omega \subset \mathbb{R}^d$, the target domain is a discrete set, $Y = \{y_1, y_2, \cdots, y_n\}$, $y_i \in \mathbb{R}^d$, the target measure is a Dirac measure, $\nu = \sum_{i=1}^n \nu_i \delta(y - y_i)$, $i = 1, 2, \ldots, n$, with the equal total mass as the source measure, $\mu(\Omega) = \sum_{i=1}^n \nu_i$. Under a *semi-discrete transport map* $T : \Omega \to Y$, a cell decomposition is induced $\Omega = \bigcup_{i=1}^n W_i$, such that every $x$ in each cell $W_i$ is mapped to the target $y_i$, $T : x \in W_i \mapsto y_i$. The map $T$ is measure preserving, denoted as $T_{\#}\mu = \nu$, if the $\mu$-volume of each cell $W_i$ equals to the $\nu$-measure of the image $T(W_i) = y_i$, $\mu(W_i) = \nu_i$. The cost function is given by $c : \Omega \times Y \to \mathbb{R}$, where $c(x, y)$ represent the cost for transporting a unit mass from $x$ to $y$. The total cost of $T$ is given by $\int_\Omega c(x, T(x)) d\mu(x) = \sum_{i=1}^n \int_{W_i} c(x, y_i) d\mu(x)$. *Semi-discrete optimal transport map* is the measure-preserving map that minimizes the total cost, $T^* := \arg\min_{T_{\#}\mu = \nu} \int_\Omega c(x, T(x)) d\mu(x)$.

When the cost function is the $L^2$ distance $c(x, y) = 1/2\|x - y\|^2$, Brenier's theorem claims that the semi-discrete OT map is given by the gradient map of a piece-wise (PL) convex function, the so-called Brenier potential $u_h : \Omega \to \mathbb{R}$, $u_h(x) := \max_{i=1}^n \{\pi_{h,i}(x)\}$, where $\pi_{h,i}(x) = \langle x, y_i \rangle + h_i$ is the hyperplane corresponding to $y_i \in Y$. As shown in Fig. 3(a), the projection of the graph of $u_h$ decomposes $\Omega$ into cells $W_i(h)$, each cell $W_i(h)$ is the projection of the supporting plane $\pi_{h,i}(x)$. The height vector $h$ is the unique optimizer of the following convex energy under the condition that $\sum_i h_i = 0$,

$$E(h) = \int_0^h \sum_{i=1}^n w_i(\eta) d\eta_i - \sum_{i=1}^n h_i \nu_i, \tag{1}$$

where $w_i(\eta)$ is the $\mu$-volume of $W_i(\eta)$. The convex energy $E(h)$ can be optimized simply by gradient descend method with $\nabla E(h) = (w_i(h) - \nu_i)^T$.

The key is to compute the $\mu$-volume $w_i(h)$ of each cell $W_i(h)$, which can be estimated using conventional Monte Carlo method. We draw $N$ random samples from $\mu$ distribution, $\{x_j\} \sim_{i.i.d.} \mu$, $\forall j \in \mathcal{J}$, the estimated $\mu$-volume of each cell is $\hat{w}_i(h) = \#\{j \in \mathcal{J} \mid x_j \in W_i(h)\}/N$. Given $x_j$, we can find $W_i$ in which $x_j \in W_i$ by $i = \arg\max_i \{\langle x_j, y_i \rangle + h_i\}$, $i = 1, 2, \ldots, n$. When $N$ is large enough, $\hat{w}_i(h)$ converges to $w_i(h)$. Then the gradient of the energy is approximated as $\nabla E \approx (\hat{w}_i(h) - \nu_i)^T$. Once the gradient is estimated, we can use Adam algorithm Kingma & Ba (2015) to minimize the energy. Sampling of $x$ is independent of each other and finding the cell that

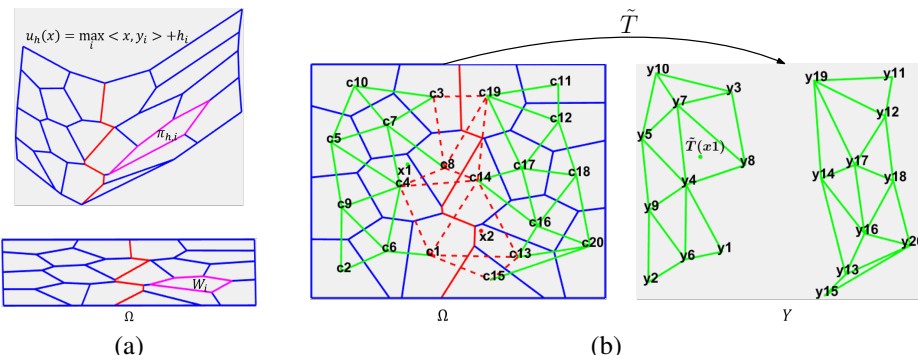

(a)                   (b)

Figure 3: Illustration of the proposed algorithm with two modes in $Y$. (a) Brenier potential and the corresponding power diagram. Each cell $W_i$ is mapped to $y_i$, which is the slope of $\pi_{h,i}$. The red line in $\Omega$ gives the singular set. (b) The extended semi-discrete map. By computing the weighted center of each cell, and then triangulating the centers according to the power diagram, we get the PL map $\tilde{T}(x)$ from $\Omega$ to $Y$. If the samples $x_1$ is not in the triangles transverse the singular set, we map it to the corresponding $\tilde{T}(x_1)$ in $Y$.

$x$ is located only involves matrix multiplication and sorting. Hence the Monte Carlo method has a natural parallel computation implementation on GPUs.

The approximation error is proportional to the inverse of the square root the amount of Monte Carlo samples. Asymptotically the number of Monte Carlo samples increases exponentially with respect to the dimension $d$ (see e.g. Weed & Bach (2017)). This brings huge computational burdens. To find a good balance between precision and speed, we *adaptively* adjust the number of random samples. In practice, we apply the following strategy: if the energy $E(h)$ ceases decreasing for a number of consecutive steps, we double the amount of Monte Carlo samples. The algorithmic details of semi-discrete OT map are summarized in Alg. 1.

---

**Algorithm 1** Semi-discrete OT Map

1: **Input:** Latent codes $Y = \{y_i\}_{i\in\mathcal{I}}$, empirical latent code distribution $\nu = \frac{1}{|\mathcal{I}|}\sum_{i\in\mathcal{I}}\delta_{y_i}$, number of Monte Carlo samples $N$, positive integer $s$.
2: **Output:** Optimal transport map $T(\cdot)$.
3: Initialize $h = (h_1, h_2, \ldots, h_{|\mathcal{I}|}) \leftarrow (0, 0, \ldots, 0)$.
4: **repeat**
5:     Generate $N$ uniformly distributed samples $\{x_j\}_{j=1}^N$.
6:     Calculate $\nabla h = (\hat{w}_i(h) - \nu_i)^T$.
7:     $\nabla h = \nabla h - mean(\nabla h)$.
8:     Update $h$ by Adam algorithm with $\beta_1 = 0.9, \beta_2 = 0.5$.
9:     **if** $E(h)$ has not decreased for $s$ steps **then**
10:         $N \leftarrow N \times 2$.
11:     **end if**
12: **until** Converge
13: OT map $T(\cdot) \leftarrow \nabla(\max_i \langle \cdot, y_i \rangle + h_i)$.

---

**Algorithm 2** Generate latent code

1: **Input:** Optimal transport map $T(\cdot)$, number of samples to generate $n$, angle threshold $\hat{\theta}$.
2: **Output:** Generated latent code $P$.
3: Compute $\hat{c}_i$ by Monte Carlo method.
4: **repeat**
5:     Sample $x \sim \mu$, Find the smallest $d+1$ vertex around $x$ as $\{d(x, \hat{c}_{i_0}), d(x, \hat{c}_{i_1}), \ldots, d(x, \hat{c}_{i_d})\}$.
6:     Compute dihedral angles $\theta_{i_k}$ between $\pi_{i_0}$ and $\pi_{i_k}$.
7:     Select $\theta_{i_k}$ with $\theta_{i_k} \leq \hat{\theta}$, result in $\hat{i}_k = 0, 1, \ldots, d_1$.
8:     **if** $\forall k, \theta_{i_k} > \hat{\theta}$ **then** Abandon $x$
9:     **else** Generate latent code $\tilde{T}(x) = \sum_{k=0}^{d_1} \lambda_k T(\hat{c}_{\hat{i}_k})$ with $\lambda_k = d^{-1}(x, \hat{c}_{\hat{i}_k})/\sum_{j=0}^{d_1} d^{-1}(x, \hat{c}_{\hat{i}_j})$.
10:     **end if**
11: **until** Generate $n$ new latent code

---

**Piece-wise Linear Extension** The semi-discrete OT map $\nabla u_h : \Omega \to Y$ maps all $x \in \Omega$ to the latent codes of training samples $\{y_i\}$'s and won't generate new samples. Therefore, we extend the semi-discrete OT map $T = \nabla u_h$ to a piecewise linear (PL) mapping $\tilde{T}$ as follows. The projection of $u_h$ in the source domain induces a cell decomposition of $\Omega$, of which each cell is of $\mu$-volume $\nu_i$ and is mapped to the corresponding $y_i$. By representing the cells by their $\mu$-mass centers as $c_i := \int_{W_i(h)} x d\mu(x)$, we can get the point-wise map $t : c_i \mapsto y_i$. The Poincaré of the cell decomposition induces a triangulation of the centers $C = \{c_i\}$: if $W_i \cap W_j \neq \emptyset$, then $c_i$ is connected with $c_j$ to form an edge $[c_i, c_j]$. Similarly, if $W_{i_0} \cap W_{i_1} \cdots \cap W_{i_k} \neq \emptyset$, then there is a $k$-dimensional simplex $[c_{i_0}, c_{i_1}, \ldots, c_{i_k}]$. All these simplices form a triangulation of $C$ (a simplicial complex), denoted as $\mathcal{T}(C)$ (the green triangles in the left of Fig. 3(b)). We can triangulate $Y$ in the same way to obtain the triangulation $\mathcal{T}(Y)$ (the green triangles in the right of Fig. 3(b)). Once a random sample $x$ is drawn from the distribution $\mu$, we can find the simplex $\sigma$ in $\mathcal{T}(C)$ containing $x$. Assume the simplex $\sigma$ has $d+1$ vertices $\{c_{i_0}, c_{i_1}, \ldots, c_{i_d}\}$, the bary-centric coordinates of $x$ in $\sigma$ is defined as $x = \sum_{k=0}^d \lambda_k c_{i_k}$, and $\sum_{k=0}^d \lambda_k = 1$ with all $\lambda_k$ non-negative. Then the generated latent code of $x$ under this piece-wise linear map is given by $\tilde{T}(x) = \sum_{k=0}^d \lambda_k y_{i_k}$ (In Fig. 3(b), the green dot $x1$ is

mapped to be $\tilde{T}(x1))$. Because all of the $y_i$s are used to construct the simplicial complex $\mathcal{T}(Y)$ in the support of the target distribution, we can guarantee that no mode is lost.

In practice, the $\mu$-mass center $c_i$ is approximated by the mean value of all the Monte-Carlo samples inside $W_i(h)$, $\hat{c}_i = \sum_{x_j \in W_i} x_j / \#\{x_j \in W_i\}$, where $x_j \sim \mu$. The connectivity information $\mathcal{T}(C)$ is too complicated to construct and to store in high dimensional space, thus $\mathcal{T}(C)$ is not explicitly built. Instead, we find the simplex $\sigma \in \mathcal{T}(C)$ containing $x$ as follows: given a random point $x \in \Omega$, evaluate and sort its Euclidean distances to the centers $d(x, \hat{c}_i), i = 1, 2, \ldots, n$ in the ascending order. Suppose the first $d+1$ items are $\{d(x, \hat{c}_{i_0}), d(x, \hat{c}_{i_1}), \ldots, d(x, \hat{c}_{i_d})\}$, then $\sigma$ is formed by $\{\hat{c}_{i_k}\}$. The bary-centric coordinates $\hat{\lambda}_{i_k}$ are estimated as $\hat{\lambda}_{i_k} = d^{-1}(x, \hat{c}_{i_k}) / \sum_{k=0}^{d} d^{-1}(x, \hat{c}_{i_k})$. However, this may generate some spurious samples. To overcome it, we need further to detect the singular set.

**Singular Set Detection** According to Figalli's theory Figalli (2010); Chen & Figalli (2017), if there are multiple modes or the support of the target distribution is concave, there will be singular sets $\Sigma \subset \Omega$, where the Brenier potential is continuous but not differentiable, making its gradient map, i.e. the transport map, discontinuous.

As shown in Fig. 3(a), the source distribution is uniformly defined on $\Omega$, and the target empirical distribution has two modes. There is one ridge (the red line) on the Brenier potential $u_h$, whose projection is the singular set $\Sigma$ (the red line in $\Omega$). $\Omega \setminus \Sigma$ consists of two connected components, each of them is mapped onto a single mode. $\Sigma$ consists of codimension 1 facets of cells. If $W_i(h) \cap W_j(h) \subset \Sigma$, then the dihedral angle between two supporting planes $\pi_{h,i}$ and $\pi_{h,j}$ of $u_h$ is prominently large. Therefore, on the graph of Brenier potential, we pick the pairs of facets whose dihedral angles are larger than a given threshold, the projection of their intersection gives a co-dimension 1 cell in the singular set $\Sigma$. During the generation process, if a random sample $x$ is around $\Sigma$, it will be mapped by $\tilde{T}$ to the gaps among the modes. When generating new latent codes, we just abandon such samples. And this help our method prevent the mode mixture phenomenon.

Given the extended OT map $\tilde{T}(x)$, some of the polyhedrons transverse the singular set (the red lines of Fig .3(b)), which means that different vertices of the polyhedron belongs to different mode. If the sample $x$ falls into such a polyhedron (the dotted red triangle), we just abandon it (as shown in Fig. 3(b), the red dot $x2$ is just abandoned). Specifically, given $x$, we can detect if it belongs to the singular set by checking the angles $\theta_{i_k}$ between $\pi_{i_0}$ and $\pi_{i_k}, k = 1, 2, \ldots, d$ as $\theta_{i_k} = \langle y_{i_0}, y_{i_k} \rangle / \|y_{i_0}\| \cdot \|y_{i_k}\|$. If all of the angles $\theta_{i_k}$ is larger than a threshold $\hat{\theta}$, we say $x$ belongs to the singular set and just abandon it. Or we just select a subset $\{\pi_{i_k}\}$ with $\theta_{i_k} \leq \hat{\theta}$, denoted as $\{\pi_{\hat{i}_k}, k = 0, 1, \ldots, d_1\}$. Then we can compute $\lambda_k = d^{-1}(x, \hat{c}_{\hat{i}_k}) / \sum_{j=0}^{d_1} d^{-1}(x, \hat{c}_{\hat{i}_j})$ and $\tilde{T}(x) = \sum_{k=0}^{d_1} \lambda_k T(\hat{c}_{\hat{i}_k})$. *Intuitively, $\tilde{T}(\cdot)$ smooths the discrete function $T(\cdot)$ in regions where latent codes are dense and keep the discontinuity of $T(\cdot)$ where latent codes are very sparse. In this way we avoid generating spurious latent code and thus improve the generation quality.* The algorithm to generate new code is shown in Alg. 2 and the effect of threshold filtering is further investigated in Appendix 4.1.

## 4 EXPERIMENTS

In order to validate that the proposed method can solve the mode collapse/mixture problems and generate controllable high quality images, several experiments are conducted.

The first experiment explores the influence of the angle thresholds for the singularity detection on synthetic dataset.

The second experiment focuses on toy sets, so that the complexity of the tasks can be manually controlled and the mode and quality of the generated samples can be accurately computed. Lin et al. (2018) did a large-scale comparison with previous methods that explicitly proposed to mitigate mode collapse and thus established a baseline for comparison. For consistent evaluation, we set up our experiment on the same benchmark dataset as theirs, and make the comparison.

In the last experiment, we run the proposed method mainly on 4 public datasets, MNIST LeCun & Cortes (2010), MNIST-FANSION Han Xiao & Vollgraf (2017), CIFAR-10 Krizhevsky (2009) and CelebA Zhang et al. (2018), just like the authors of Hoshen & Malik (2019) Sajjadi et al. (2018) Lucic et al. (2018) did in their papers. Besides, the architecture of the decoder is the same as Lucic et al. (2018), in which the authors did a large-scale study to evaluated the best performance of 8

different generative models including various GAN models and VAE, and the encoder is set to be the mirror of decoder.

## 4.1 SINGLE PARAMETER SELECTIVE INTERPOLATION

On synthetic datasets, effects of angle threshold filtering can be visually inspected. As illustrated in Fig. 4, number of mode is a monotonically increasing function with respect to angle threshold $\hat{\theta}$. Quality of generated samples is effected directly by choosing different $\hat{\theta}$. Generally, small $\hat{\theta}$ encourages interpolation in between closely related real samples while too large $\hat{\theta}$ will result in interpolation between samples from different modes, which might in turn lower generation quality. On synthetic datasets, where modes are isotropic and different modes are clearly separable, an ideal $\hat{\theta}$ that captures all modes while avoids generating low quality samples can be chosen within a relatively wide band. For real world datasets of unknown modes, an ideal $\hat{\theta}$ needs to be hand tuned as the separability of different modes depends largely on input data pattern and quality of the embedding map.

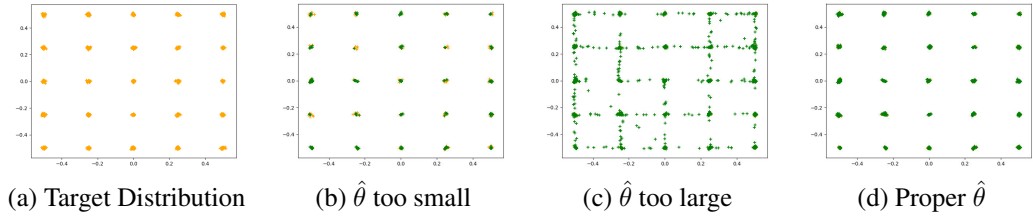

(a) Target Distribution     (b) $\hat{\theta}$ too small     (c) $\hat{\theta}$ too large     (d) Proper $\hat{\theta}$

Figure 4: Effect of increasing angle threshold $\hat{\theta}$. (a) shows target distribution. (b) and (c) shows AE-OT results when $\hat{\theta}$ is too small (as in (b)) or too large (as in (c)). (d) shows a proper choice of $\hat{\theta}$ that precisely captured and generalized all modes.

## 4.2 MITIGATION OF MODE COLLAPSE AND MODE MIXTURE IN SYNTHETIC DATASET

Since synthetic dataset consists of explicit distributions and known modes, mode collapse and the quality of the generated sample can be accurately measured. We choose the same synthetic datasets and metrics as in Lin et al. (2018). Specifically, we use 2D-ring and 2D-grid for test sets and *Number of modes*, *Percentage of high-quality samples*, *reverse Kullback-Leibler (KL) divergence* as evaluation metrics. *Number of modes* counts the amount of modes captured by samples produced a generative model. *Percentage of high-quality samples* measures the proportion of samples that generated within three standard deviations of the nearest mode. *reverse KL divergence* measures how well generated samples balance among all modes regarding the real distribution. In Lin et al. (2018), the authors evaluated GAN, ALI, MD and PacGAN on synthetic sets with above three metrics. Each networks are trained under the same generator architecture with a total of approximated 400K training parameters. For AE-OT test, since the source space and the target space are both 2-dimensional, there is no need to train any autoencoder. A two dimensional extended OT map is directly computed. Our results are included in table 1, and benchmarks of previous methods are copied from Lin et al. (2018) and Xiao et al. (2018). Generally speaking, the samples generated by the proposed method can not only cover all of the modes, the quality of them is also better than others.

Besides, we experiment on stack MNIST dataset and CelebA dataset to further illustrate the performance of the proposed method, and the results are shown in Section C.1 and C.2 in the Appendix.

## 4.3 QUANTITATIVE COMPARISON WITH FID

FID is computed by: (1) extract the visual-meaningful features of both the generated and real images through the inception network, (2) fit the features in both the generated and real feature spaces with Gaussian distribution, and then (3) compute the distance between the two Gaussian distributions with the following formula $FID = \|\mu_r - \mu_g\|_2^2 + Tr(\Sigma_r + \Sigma_g - 2(\Sigma_r \Sigma_g)^{\frac{1}{2}})$, where $\mu_r, \mu_g$ mean the means of the real and generated features, $\Sigma_r, \Sigma_g$ represent the variances of both distributions.

We report our results in Tab. (2), in which the compared data comes from Lucic et al. (2018)Hoshen & Malik (2019). In general, the proposed model achieves better than or comparable scores to other

Table 1: Experiments on synthetic datasets. Under standard benchmark settings, AE-OT achieves best performances over an average of 10 independent experiment results in terms of modes captured, probability of high quality samples and reverse KL divergence. The mean values and standard deviations of the experiment results are reported here.

| | 2D-ring | | | 2D-grid | | |
|---|---|---|---|---|---|---|
| | Modes (Max 8) | high quality samples | reverse KL | Modes (Max 25) | high quality samples | reverse KL |
| GAN | 6.3±0.5 | 98.2±0.2% | 0.45±0.09 | 17.3±0.8 | 94.8±0.7% | 0.70±0.07 |
| ALI | 6.6±0.3 | 97.6±0.4% | 0.36±0.04 | 24.1±0.4 | 95.7±0.6% | 0.14±0.03 |
| MD | 4.3±0.8 | 36.6±8.8% | 1.93±0.11 | 23.8±0.5 | 79.9±3.2% | 0.18±0.03 |
| PacGAN2 | 7.9±0.1 | 95.6±2.0% | 0.07±0.03 | 23.8±0.7 | 91.3±0.8% | 0.13±0.04 |
| PacGAN3 | 7.8±0.1 | 97.7±0.3% | 0.10±0.02 | 24.6±0.4 | 94.2±0.4% | 0.06±0.02 |
| PacGAN4 | 7.8±0.1 | 95.9±1.4% | 0.07±0.02 | 24.8±0.2 | 93.6±0.6% | 0.04±0.01 |
| BourGAN | 8.0±0.0 | 99.8±2.9% | 4e-4±2e-4 | 24.9±0.3 | 95.9±0.2% | 0.01±0.02 |
| AE-OT | 8.0±0.0 | 99.6±0.3% | 0.004±0.001 | 25.0±0.0 | 99.8±0.2% | 0.007±0.002 |

Table 2: Quantitative comparison with FID

| | Adversarial | | | | Non-Adversarial | | Reference | |
|---|---|---|---|---|---|---|---|---|
| Dataset | NS GAN | LSGAN | WGAN | BEGAN | VAE | GLANN | AE | Ours |
| MNIST | 6.8±0.5 | 7.8±0.6 | 6.7±0.4 | 13.1±1.0 | 23.8±0.6 | 8.6±0.1 | 5.5 | **6.2±0.2** |
| Fansion | 26.5±1.6 | 30.7±2.2 | 21.5±1.6 | 22.9±0.9 | 58.7±1.2 | 13.0±0.1 | 4.7 | **10.1±0.3** |
| CIFAR-10 | 58.5±1.9 | 87.1±47.5 | 55.2±2.3 | 71.4±1.6 | 65.4±0.2 | 46.5±0.2 | 28.2 | **38.3±0.5** |
| CelebA | 55.0±3.3 | 53.9±2.8 | 41.3±2.0 | **38.9±0.9** | 85.7±3.8 | 46.3±0.1 | 67.5 | 68.4± 0.5 |

state-of-the-art generative models. Theoretically, the FID scores of our proposed generative models should be close to that of the pre-trained autoencoders, and this is also validated in our experiments.

The autoencoder architecture we use here cannot find a good encoding for the CelebA dataset due to the limited capacity. But the FID score of the generation model is still approach to the autoencoder. In order to verify that with appropriate capacity of autoencoder, the proposed model works. We use the generator of DCGAN Radford et al. (2016) as the decoder of the autoencoder, then the reported FID score is 28.6, outperforming other models. Also, some of the generated images are displayed in Fig. 7 of the Appendix.

We also display the generating results for the four dataset in Fig. 5. It includes the original images, the best generating results of Lucic et al. (2018), including various GANs and VAE, the results of Hoshen & Malik (2019) and ours, row by row.

## 5 CONCLUSION

This work gives a theoretic explanation for mode collapse/mixture by Brenier's theory and Figalli's regularity theory of optimal transport maps. When the target measure has concave support, the OT map is discontinuous on the signular sets. But DNNs can only represent continuous functions, this conflict causes the both problems. In order to solve this problem, the AE-OT model is proposed by separating manifold embedding and measure transformation. The former step is computed using an autoencoder, the latter is carried out using the extended semi-discrete OT map based on GPUs. The model is tested thoroughly and extensively by both synthetic and real data sets. The experimental results validates the discontinuity of the OT maps and demonstrate the advantages comparing to the state-of-the-arts.

**Acknowledgements**

The project is partially supported by the National Natural Science Foundation of China (61936002, 61772105, 61720106005), NSF CMMI-1762287 collaborative research: computational framework for designing conformal stretchable electronics, Ford URP topology optimization of cellular mesostructures' nonlinear behaviors for crash safety, and NSF DMS-1737812 collaborative research: ATD: theory and algorithms for discrete curvatures on network data from human mobility and monitoring.

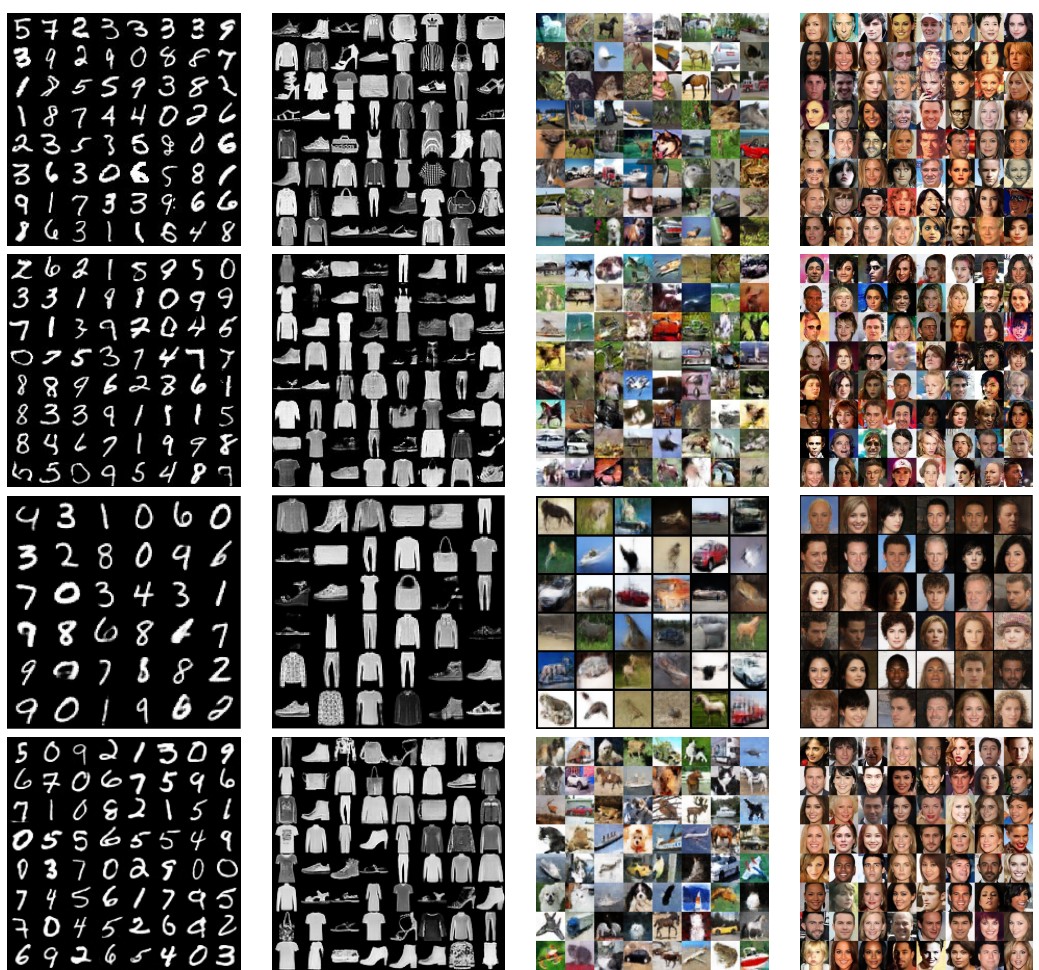

Figure 5: The comparison between the proposed method and the state-of-the-art on MNIST LeCun & Cortes (2010), Fashion MNIST Han Xiao & Vollgraf (2017), Cifar10 Krizhevsky (2009) and CelebA Zhang et al. (2018). The first row shows some of the real images in each dataset. The second row corresponds to the best results of Lucic et al. (2018); The third row gives the results of Hoshen & Malik (2019); then we display our generating results in the last row.

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

# A  BRENIER'S THEORY

In this subsection, we briefly introduce the basic concepts and theorems in optimal transport theory, which comes from Brenier theory Villani (2008); Brenier (1987; 1991a), and discrete theory Gu et al. (2016).

**Optimal transport Problem**    Suppose $X, Y \subset \mathbb{R}^d$ are two subsets of $n$-dimensional Euclidean space, $\mu, \nu$ are two probability measured defined on $X$ and $Y$ respectively, with equal total measure, $\mu(X) = \nu(Y)$. A map $T : X \rightarrow Y$ is *measure preserving*, denoted as $T_{\#}\mu = \nu$, if for any measurable set $B \subset Y$, $\mu(T^{-1}(B)) = \nu(B)$. Given a cost function $c(x, y) : X \times Y \rightarrow \mathbb{R}_{\geq 0}$, indicating the cost of moving each unit mass from the source to the target, the total *transport cost* of the map $T$ is defined to be $\int_X c(x, T(x)) d\mu(x)$.

The *Monge's problem* of optimal transport arises from finding the measure-preserving map that minimizes the total transport cost.

$$(MP) \quad \mathcal{W}_c(\mu, \nu) := \min_{T_{\#}\mu = \nu} \int_X c(x, T(x)) d\mu(x). \tag{2}$$

The solutions to the Monge's problem is called the *optimal transport map*, whose total transport cost is called the *Wasserstein distance* between $\mu$ and $\nu$, denoted as $\mathcal{W}_c(\mu, \nu)$.

**Brenier's Approach**    Brenier Brenier (1987; 1991a) proved the following theorem:

**Theorem 1** (Brenier Brenier (1987; 1991b)). *Suppose $X$ and $Y$ are the Euclidean space $\mathbb{R}^d$ and the transport cost is the quadratic Euclidean distance $c(x, y) = 1/2\|x - y\|^2$. Furthermore $\mu$ is absolutely continuous and $\mu$ and $\nu$ have finite second order moments, then there exists a convex function $u : X \rightarrow \mathbb{R}$, the so-called Briener potential, its gradient map $\nabla u$ gives the solution to the Monge's problem. The Brenier potential is unique up to a constant.*

Brenier's polar factorization theorem claims that: for any measure preserving map $T_{\#}\mu = \nu$, $T$ can be uniquely decomposes into the forms $T = \nabla u \circ s$, where $s : X \rightarrow X$ is a volume preserving map and $\nabla u$ is the optimal transport map under $L^2$ cost. Therefore, the regularity of $T$ can be determined by that of $\nabla u$.

**Discrete Brenier's Theorem**    Brenier theorem can be directly generalized to discrete target measure. Suppose the source measure $\mu$ is defined on a compact convex set $\Omega$, the target measure $\nu = \sum_{i=1}^n \nu_i \delta(y - y_i)$, $\mu(\Omega) = \sum_i \nu_i$. The discrete Brenier potential is a piecewise linear function,

$$u_h(x) = \max_{i=1}^n \{\pi_{h,i}(x)\} = \max_{i=1}^n \{\langle x, y_i \rangle + h_i\}. \tag{3}$$

As shown in Fig. 3(a), the projection of the Brenier potential induces a cell decomposition of $\Omega$, each cell $W_i(h) := \{p \in \Omega | \nabla u_h(p) = y_i\}$, whose $\mu$-measure is denoted as $w_i(h)$.

**Theorem 2** (Discrete Brenier Theorem Gu et al. (2016)). *For any $\nu_1, \nu_2, \ldots, \nu_n > 0$ with $\sum_{i=1}^n \nu_i = \mu(\Omega)$, there exists $h = (h_1, h_2, \ldots, h_n) \in \mathbb{R}^n$, unique up to adding a constant $(c, c, \ldots, c)$, so that $w_i(h) = \nu_i$, for all $i$. The vector $h$ is the unique minimum argument of the following convex energy*

$$E(h) = \int_0^h \sum_{i=1}^n w_i(\eta) d\eta_i - \sum_{i=1}^n h_i \nu_i, \tag{4}$$

*defined on an open convex set $\mathcal{H} = \{h \in \mathbb{R}^n : w_i(h) > 0, i = 1, 2, \ldots, n\}$. Furthermore, $\nabla u_h$ minimizes the quadratic cost $\int_\Omega \|x - T(x)\|^2 d\mu(x)$ among all transport maps $T_{\#}\mu = \nu$. The gradient of above energy is given by $\nabla E(h) = (w_1(h) - \nu_1, w_2(h) - \nu_2, \ldots, w_n(h) - \nu_n)^T$. The Hessian of the energy is given by*

$$\frac{\partial w_i}{\partial h_j} = -\frac{\mu(W_i \cap W_j)}{\|y_i - y_j\|}, \frac{\partial w_i}{\partial h_i} = \sum_{j \neq i} \frac{\partial w_i}{\partial h_j}. \tag{5}$$

The optimal transport map can be obtained by convex optimization. Furthermore, the optimization can be carried out using Newton's method. The global linear convergence rate is guaranteed by the following theorem:

**Theorem 3** (Kitagawa-Mérigot-Thibert Kitagawa et al. (2019)). *Assume the cost function is quadratic distance, $\mu$ has convex support and also that (i) The probability density of $\mu$ is $C^{0,\alpha}(\Omega)$ for $\alpha$ in $(0; 1]$. (ii) $\mu$ has positive Poincaré-Wirtinger constant. Then the Newton algorithm for semi-discrete optimal transport converges globally with linear rate and locally with rate $1 + \alpha$.*

Though in our method the gradient descend method is applied, the above theorem also ensures its convergence because of the convexity of the energy function $E(h)$ we adopted.

## B  FIGALLI'S THEORY

In this section, we show the fact that even for the case of **single mode**, the transport map may still be **discontinuous**, which will cause the instability of the training process of GANs. The arguments are mainly based on the regularity theory of transport maps developed by Figalli Chen & Figalli (2017); Figalli (2010) and so on.

According to Brenier's Theorem Brenier (1987; 1991a), any transport map can be decomposed into a measure preserving map and a solution to the Monge-Ampére equation, which is the optimal transport map under the $L^2$ cost function. Therefore, the continuity of the transport map can be reduced to the regularity (smoothness) of the solution to the Monge-Ampére equation. When the support of the target measure is convex and the density functions are smooth, Caffarelli showed the map is differentiable; otherwise if the target domain is not convex, Figalli showed the map is discontinuous, and gave precise description of the singularity set. In this section, we briefly introduce Figallis' theory, and conduct an experiment using CelebA data set to show the existence of the singularity set, hence demonstrate the fact that the transport maps computed in GANs are discontinuous.

### B.1  CONVEX DOMAINS - CAFFARELLI THEOREM

Let $\Omega$ and $\Lambda$ are two bounded open sets in $\mathbb{R}^n$, and let $f : \mathbb{R}^n \to \mathbb{R}$ and $g : \mathbb{R}^n \to \mathbb{R}$ be two positive functions such that $f = 0$ in $\mathbb{R}^n \setminus \Omega$, $g = 0$ in $\mathbb{R}^n \setminus \Lambda$, and

$$\int_\Omega f = \int_\Lambda g = 1.$$

According to Brenier's Theorem Brenier (1987; 1991a), there exists a globally Lipschitz convex function $\varphi : \mathbb{R}^n \to \mathbb{R}$ such that $\nabla \varphi_{\#} f = g$ and and $\nabla \varphi(x) \in \bar{\Lambda}$ for $\mathcal{L}^2$-a.e. $x \in \mathbb{R}^n$. We say $\varphi$ weakly solves the Monge-Ampére equation

$$det(D^2\varphi) = \frac{f}{g \circ \nabla\varphi} \quad in \ \mathbb{R}^n, \tag{6}$$

together with the boundary condition $\nabla\varphi(\mathbb{R}^n) \subset \bar{\Lambda}$. $\varphi$ is called the *Briener potential*.

As shown by Caffarelli [9], if $\Lambda$ is convex, then $\varphi$ is strictly convex, and it solves the Monge-Ampére equation 6. The regularity theory has been estabilished (see Caffarelli (1990a;b; 1991)), such as

1. if $\lambda \leq f, g \leq 1/\lambda$ for some $\lambda > 0$, then $\varphi \in C^{1,\alpha(\lambda)}_{loc}(\Omega)$.
2. if $f \in C^{k,\alpha}_{loc}(\Omega)$ and $g \in C^{k,\alpha}_{loc}(\Lambda)$, then $\varphi \in C^{k+2,\alpha}_{loc}(\Omega)$, $(k \geq 0, \alpha \in (0,1))$.

### B.2  NON-CONVEX DOMAINS - FIGALLI THEOREM

However, if $\Lambda$ is not convex, the regularity of the Brenier potential can not be guaranteed. Figalli gave examples in Figalli (2010), such that

1. $\Omega$ is convex, $\Lambda$ is simply connected, but non-convex;
2. the density functions $f$ and $g$ are smooth, $f \in C^\infty(\Omega)$ and $g \in C^\infty(\Lambda)$;
3. the Brenier potential $\varphi \notin C^1(\Omega)$, the transport map $\nabla\varphi$ is not continuous.

In this scenario, the transport map can not be learned using DNNs, and training process is unstable or the GAN model generates unrealistic samples.

**Figalli's construction** Let $\varphi : \mathbb{R}^n \to \mathbb{R}$ be a convex function. Its *subdifferential* at a point $x$ is defined by

$$\partial \varphi(x) := \{y \in \mathbb{R}^n | \varphi(z) \geq \varphi(x) + y \cdot (z - x), \forall z \in \mathbb{R}^n\}.$$

$\varphi$ is differentiable at a point $x$ if and only if $\partial \varphi(x)$ is a singleton. Figalli decomposes the set of non-differentiability points according to the dimension of the singular set:

$$\Sigma_k(\varphi) := \{x \in \mathbb{R}^n | dim(\partial \varphi) = k\}, k = 0, \ldots, n. \tag{7}$$

For any $k = 0, \ldots, n$, the set $\Sigma_k(\varphi)$ is $(n - k)$-rectifiable. The set of *reachable subgradients* at $x$ as

$$\nabla_* \varphi := \left\{ \lim_{k \to +\infty} \nabla \varphi(x_k) | x_k \in \Sigma_0, x_k \to x \right\}.$$

It is known that the convex hull of $\nabla_* \varphi(x)$, coincides with $\partial \varphi(x)$.

**Theorem 4** (Figalli). *Assume that there exists $\lambda > 0$ such that $\lambda \leq f \leq 1/\lambda$ in $\Omega$, $\lambda \leq g \leq 1/\lambda$ in $\Lambda$, and that $\partial \Omega$ and $\partial \Lambda$ are continuous. Then $\varphi$ is strictly convex inside $\Omega$. Moreover there exist two open sets $\Omega' \subset \Omega$ and $\Lambda' \subset \Lambda$, with $\mathcal{L}^2(\Omega \setminus \Omega') = \mathcal{L}^2(\Lambda \setminus \Lambda') = 0$, such that $\varphi \in C^{1,\alpha}(\Omega')$, $\nabla \varphi$ is a bi-Hölder homeomorphism between $\Omega'$ and $\Lambda'$, and $\varphi$ is an Alexandrov solution of 6 inside $\Omega'$. In particular, Caffarelli's regularity theory for strictly convex Alexandrov solutions of the Monge-Ampére equations applies to $\varphi$ inside $\Omega'$.*

Figalli studies the singular set of $\varphi$ in $\Omega$, i.e. the set of points $x \in \Omega$ where $\varphi$ is not differentiable, denoted as $Sing$. Figalli shows the singularity set has the following characterization,

$$Sing = \{x \in \Omega | \partial \varphi(x) \cap \Lambda = \emptyset, \nabla_* \varphi(x) \subset \partial \Lambda, \partial \varphi(x) \not\subset \Lambda\}.$$

it can be decomposed into connected components $Sing := \cup_i S_i$. For planar case,

**Theorem 5** (Figalli Singularity Set). *The number of connected components of $Sing$ is at most countable. Moreover:*

1. *either $S_i$ coincides with an isolated point $\{xi\}$ for some $x_i \in \Omega$, and in this case the boundary of $\partial(xi)$ is entirely contained inside $\partial \Lambda$ (so that $\partial \varphi(xi)$ completely fills a hole in $\Lambda$);*

2. *or $S_i$ can be written as a disjoint union as follows:*

$$S_i = \bigcup_j \gamma_{ij},$$

*where $\gamma_{ij} : I_{ij} \to Sing$ are embedded Lipschitz curves parameterized by arc-lengh, $I_{ij}$ is an interval.*

### B.3 ELEMENTARY EXPERIMENTS

We have designed a numerical experiment to verify Figalli's theorems in low dimensional case.

Fig. 6 shows another computational result, which demonstrates the singularity structure in Figalli's theorem. The source domain $\Omega$ is the unit disk, the target domain $\Lambda$ is with complicated geometry. The singularity set of the optimal transport map satisfies the description of Figalli's theorem 4,

$$\Sigma_1 = \bigcup_{i=0}^{3} \gamma_k, \ \Sigma_2 = \bigcup_{j=0}^{1} x_j.$$

$\partial \varphi(x_0)$ fills the hole on $\Lambda$. For any interior point $p \in \gamma_1$, $\partial \varphi(p)$ is a line segment connecting two points on the boundary of $\Lambda$.

## C ADDITIONAL EXPERIMENTS

### C.1 STACK MNIST EXPERIMENT

Experiments of varisou GAN models on stacked MNIST dataset are in consistent with Lin et al. (2018). For AE-OT model, we use the architecture shown in table 3 and table 4, with the decoder

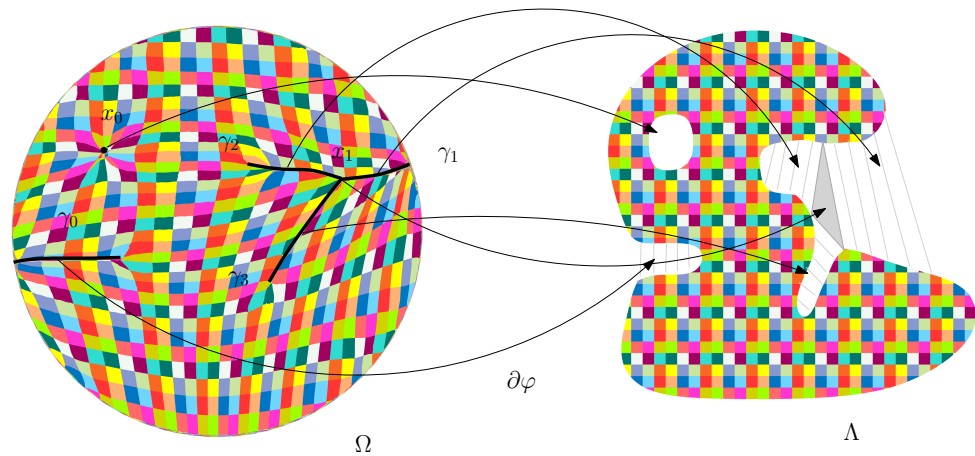

Figure 6: Figalli's example: Singularity structure of an optimal transport map, Fig.3.2 in Figalli (2010), Lei et al. (2020).

Table 3: Encoder architecture for stack MNIST

| layer | number of outputs | kernel size | stride | BN | activation |
|---|---|---|---|---|---|
| Input $x \sim P_{data}$ | 28*28*3 | | | | |
| Convolution | 14*14*16 | 5*5 | 2 | | LeakyReLU |
| Convolution | 7*7*32 | 5*5 | 2 | Yes | LeakyReLU |
| Convolution | 4*4*64 | 5*5 | 2 | Yes | LeakyReLU |
| Convolution | 2*2*128 | 5*5 | 2 | Yes | LeakyReLU |
| Fully connected | 100 | | | | |

architecture same as the consistent generator architectures in GANs, and encoder having mirrored architecture.

We test diversity of generated samples from our AE-OT method on stack MNIST dataset that consists of 128,000 samples in 1,000 modes with each sample stacking three handwritten digit images from MNIST dataset LeCun et al. (1998). *Number of modes* counts the amount of modes captured by samples produced a generative model. The *reverse KL divergence* is computed by first assign each samples to their nearest mode, and compute the KL divergence between histogram of sample count on each mode and the histogram of real data. We choose angle threshold $\hat{\theta} = 0.5$ for AE-OT method. Details of network architectures are listed in supplementary materials. Experiments results are summarized in table 5, which show our method achieves best performance in terms of modes captured and reverse KL divergence on stacked MNIST dataset.

## C.2 CELEBA EXPERIMENT

we evaluate our method on CelebA dataset by measuring collision probability in a batch of 1024 generated images of size 64-by-64. If a pair of identical images appear, a collision is declared, and thus higher collision probability means lower generation diversity. The same metric has been used in Lin et al. (2018) for evaluation of PacGAN. To make a consistent comparison, we design our autoencoder network with encoder having the same architecture as in previous work and decoder having a mirrored architecture of encoder. Angle threshold $\hat{\theta}$ is chosen to be 0.7 for AE-OT test. Results are listed in table 6, with corresponding images can be downloaded here. Results have shown that our method achieves best result in terms of probability of collision. Autoencoder network structures can be found at table 7 and 8.

Table 4: Decoder architecture for stack MNIST

| layer | number of outputs | kernel size | stride | BN | activation |
|---|---|---|---|---|---|
| Input $z \sim P_{latent}$ | 100 | | | | |
| Fully connected | 2*2*128 | | | Yes | ReLU |
| Transposed Convolution | 4*4*64 | 5*5 | 2 | Yes | ReLU |
| Transposed Convolution | 7*7*32 | 5*5 | 2 | Yes | ReLU |
| Transposed Convolution | 14*14*16 | 5*5 | 2 | Yes | ReLU |
| Transposed Convolution | 28*28*3 | 5*5 | 2 | | Tanh |

Table 5: Experiments on stacked MNIST. Results have shown that our method achieves best results in terms of mode captured and reverse KL divergence. (*) In WGAN, PacWGAN and AE-OT experiments, number of feature maps in each network layer is a quarter of those in other experiments.

| | Stacked MNIST | |
|---|---|---|
| | Modes | KL |
| DCGAN | 99.0 | 3.40 |
| ALI | 16.0 | 5.40 |
| Unrolled GAN | 48.7 | 4.32 |
| VEEGAN | 150.0 | 2.95 |
| MD | $24.5 \pm 7.67$ | $5.49 \pm 0.42$ |
| PacDCGAN4 | $1000.0 \pm 0.00$ | $0.07 \pm 0.005$ |
| WGAN(*) | $314.3 \pm 38.54$ | $2.44 \pm 0.170$ |
| PacWGAN4(*) | $965.7 \pm 19.07$ | $0.42 \pm 0.094$ |
| AE-OT(*) | $\mathbf{1000.0 \pm 0.0}$ | $\mathbf{0.03 \pm 0.0008}$ |

### C.3 LINEAR INTERPOLATION IN THE LATENT SPACE

Given any two images in the dataset, we can find the images between them by linear interpolation in the noise space because the one to one correspondence between $\mu$ mass centers in the noise space and the images in the dataset is provided by the proposed algorithm. For other generation models, though the interpolation can be done successfully in the noise space, they cannot find the correspondence from the noise space and the image space. The results of the linear interpolation are shown in Fig. 8.

Table 6: Probability of identical images in a batch of 1024 generated images from DCGAN, PacGAN2 and AE-OT. Results have shown that our method achives best result in terms of collision probability on CelebA dataset.

| Discriminator size | Probability of collision | | |
|---|---|---|---|
| (Decoder size) | DCGAN | PacDCGAN2 | AE-OT |
| 273K | 1 | 0.33 | **0** |
| 4×273K | 0.42 | 0 | **0** |
| 16×273K | 0.86 | 0 | **0** |
| 25×273K | 0.65 | 0.17 | **0** |

Table 7: Encoder architecture in CelebA experiment

| layer | number of outputs | kernel size | stride | BN | activation |
|---|---|---|---|---|---|
| Input $x \sim P_{data}$ | 64*64*3 | | | | |
| Convolution | 32*32*dim_f | 4*4 | 2 | | LeakyReLU |
| Convolution | 16*16*dim_f*2 | 4*4 | 2 | Yes | LeakyReLU |
| Convolution | 8*8*dim_f*4 | 4*4 | 2 | Yes | LeakyReLU |
| Convolution | 4*4*dim_f*8 | 4*4 | 2 | Yes | LeakyReLU |
| Convolution | 100 | 4*4 | 1 | | |

Table 8: Decoder architecture in CelebA experiment

| layer | number of outputs | kernel size | stride | BN | activation |
|---|---|---|---|---|---|
| Input $z \sim P_{latent}$ | 100 | | | | |
| Transposed Convolution | 4*4*dim_f*8 | | | | |
| Transposed Convolution | 8*8*dim_f*4 | 4*4 | 2 | Yes | ReLU |
| Transposed Convolution | 16*16*dim_f*2 | 4*4 | 2 | Yes | ReLU |
| Transposed Convolution | 32*32*dim_f | 4*4 | 2 | Yes | ReLU |
| Transposed Convolution | 64*64*3 | 4*4 | 2 | | Tanh |

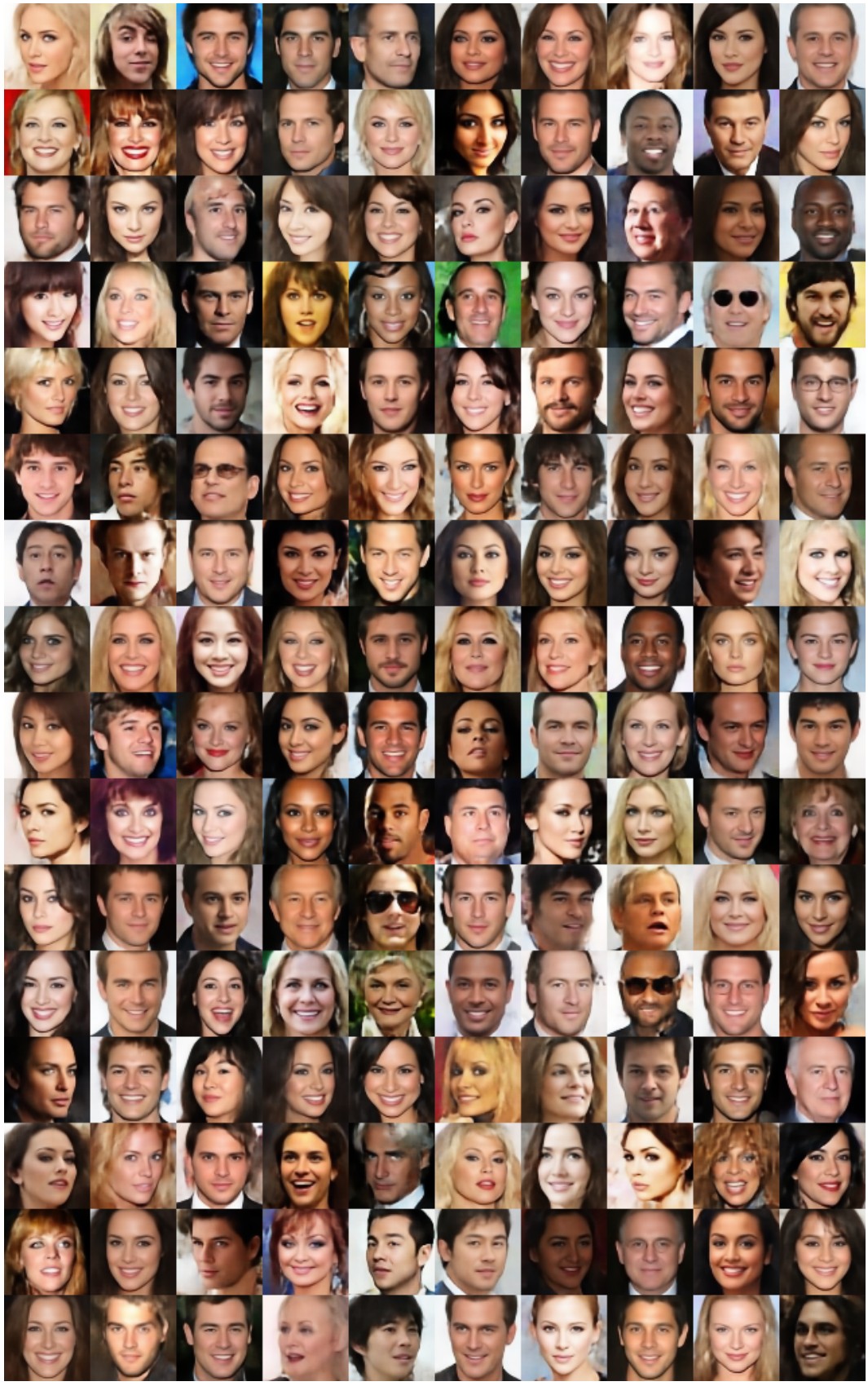

Figure 7: The generated human faces with the architecture originated from DCGAN Radford et al. (2016)

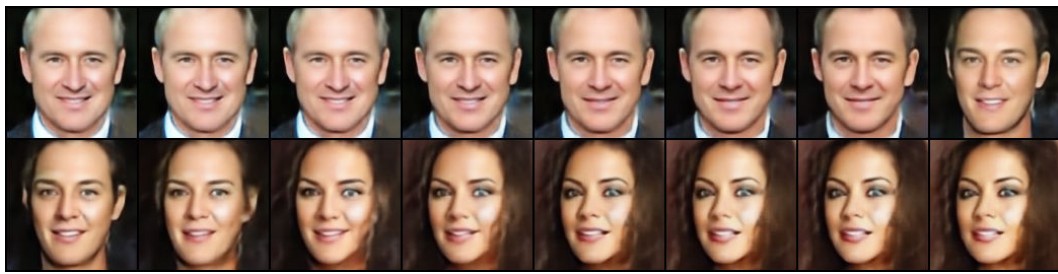

Figure 8: The linear interpolation between given two faces in the dataset.

