# OpenReview forum: "AE-OT: A NEW GENERATIVE MODEL BASED ON EXTENDED SEMI-DISCRETE OPTIMAL TRANSPORT"
_ICLR.cc/2020/Conference — Accept (Poster)_

### Official Review · AnonReviewer1 · 2019-10-25
**Official Blind Review #1**

**Rating:** 3

**Review:**

Contributions:
1. This paper proposes a new problem in GAN distribution mapping: the concavity of support problem.
2. This paper provides a solution to the concave support together with mode collapsed problem in GAN, via a discrete-continuous optimal transport model, given some post-processing techniques to rule out "singular points".
3. Empirical results show the effectiveness of the proposed method.

To summarize their method. First, they fit a good auto-encoder model to get embeddings for the observed data as an empirical distribution \nu on space Z. Second, they use a semi-discrete OT to map a noise distribution \mu to \nv. Since OT will be aware of all modes in \nu, singular points can be detected by checking the angle between "shards" and those points that are around the "ridge" should be rejected. Thus, the proposed method could handle both the concave support problem and the mode collapse problem.

My concern is whether the proposed method is overkill because the singular point detection can be very tricky and relies on heavy linear programming. Could you explain why not using the following substitute:
Step 1. Fit an auto-encoder just as you did in the paper and get an empirical distribution \nu.
Step 2. Fit a Gaussian mixture model on \nu and do model selection over # clusters.
Step 3. Sample from the Gaussian mixture model to generate fresh images.

Since this method relies on a high-quality auto-encoder model, it is hard to say this paper makes progress in fixing the GAN's mode collapsed problem. Besides, the paper does not involve an adversarial training module. So I will not treat it as a satisfactory improvement over GAN. Overall, the proposed problem in GAN indeed exists. But the solution seems to deviate from the goal the paper aim to achieve.

**Experience Assessment:**

I have read many papers in this area.

**Review Assessment: Checking Correctness Of Derivations And Theory:**

I assessed the sensibility of the derivations and theory.

**Review Assessment: Checking Correctness Of Experiments:**

I carefully checked the experiments.

**Review Assessment: Thoroughness In Paper Reading:**

I read the paper thoroughly.

---

> ### Author Response · Authors · 2019-11-09
> **Response to Reviewer #1**
>
>
>
> ----------------------------
> Q1: My concern is whether the proposed method is overkill because the singular point detection can
> be very tricky and relies on heavy linear programming.
>
> Answer: The detection of singularities is direct and simple, for the convex polyhedron of the Brenier
> potential, just compute the inner product of the normals to each pair of adjacent facets. If the inner
> product is too big, then the projection of the intersection between the facets is in the singularity set.
> So this work doesn’t involve any linear programming at all.
>
> ----------------------------
> Q2: Could you explain why not using the following substitute: Step 1. Fit an auto-encoder just as
> you did in the paper and get an empirical distribution ν. Step 2. Fit a Gaussian mixture model on ν
> and do model selection over # clusters. Step 3. Sample from the Gaussian mixture model to generate
> fresh images.
>
> Answer: We thank Reviewer #1 for the suggestions. The proposed approach is inspiring, but it has
> potential drawbacks:
> • If the empirical distribution has only one mode, but the support is concave, then the proposed
> method still can not avoid generating unrealistic samples.
> • If the empirical distribution has multiple modes, the resulting Gaussian mixture will fill
> the gaps among the modes, therefore the proposed method still can not avoid generating
> unrealistic samples (mode mixture).
> • Fitting Gaussian mixture itself is expensive and without further assumptions, the convergence
> of the GMM fitting cannot be guaranteed.
>
> In order to show the above claims, we did the following experiments: firstly we fit the 60K latent code
> of MNIST dataset by GMM, with the number of modes set to be 10, 30, 100. Then t-SNE is used to
> visualize the data. The blue crosses are the generated data by the GMM model and the green circles
> are the training data. In the anonymous website https://drive.google.com/file/d/12HbiQNAoTpxnk-h10LY0O90j8QhSIKqw/view?usp=sharing, we provide the results: Fig. (a)(b)(c) show the generation results of GMM,
> from which we can see that there are huge number of generated samples in the regions among the
> modes. While for the proposed method, as shown in Fig. (d), nearly no generated samples fall into
> the gaps.
>
> ----------------------------
> Q3: Since this method relies on a high-quality auto-encoder model, it is hard to say this paper
> makes progress in fixing the GAN’s mode collapsed problem. Besides, the paper does not involve an
> adversarial training module. So I will not treat it as a satisfactory improvement over GAN. Overall,
> the proposed problem in GAN indeed exists. But the solution seems to deviate from the goal the
> paper aim to achieve.
>
> Answer: The real goal of this work is to tackle mode collapse and mode mixture problems in general generative models, not only for GANs. Our work targets at analysis and improvement of generators in generic generative models, including VAEs and GANs. In fact, generators in these models tend to map a unimodal Gaussian to the complex data distribution, which will inevitably encounter the singularity problem proposed in our work. We thank the reviewer #1 for pointing out the ambiguity of our motivation. We have revised our abstract and introduction parts, which illustrate that the proposed AE-OT model solves the discontinuity problems encountered by both GANs and VAEs. Actually, in the original version of our paper, we have reviewed all the DNN based generative models in the related work part, and made comparisons with GANs, VAEs and other generated models in the experiment part.
> According to Figali’s Fields medal work, it shows the intrinsic reason for mode collapse is the discontinuity of transportation map, caused by the concavity of the support of the data distributions. Based on this theoretic discovery, the AE-OT model is proposed. This model is not a conventional GAN model, but a novel generative model that exactly solves the main problems we are targeting at.

---

### Official Review · AnonReviewer3 · 2019-10-26
**Official Blind Review #3**

**Rating:** 8

**Review:**

General Comments:  The generator in Generative Adversarial Networks (GANS) computes an optimal transportation from the noise distribution to the data distribution.  However, such maps are in general discontinuous.  Since deep neural networks can only represent continuous maps, this brings two problems: mode collapse and mode mixture. This paper approaches both problems using Figalli's regularity theory. They separate the manifold embedding (here an autoencoder maps input data to a latent space) from the optimal transportation (this map is found by convex optimization). Composing these two steps yields the proposed method. Their method basically avoids representing discontinuous maps by the generator. Empirically, the proposed method performs similar or better than state-of-the-art.

I think the idea of the paper is nice, and an interesting perspective  on GANs is presented. A new method is proposed. The numerical contributions are certainly significant. Therefore, I believe the paper deserves publication.

Nevertheless, I have some comments below.

1) Although this paper brings a new perspective, based on optimal transport theory, as far as I can understand this paper does not establish formal new results. Thus I think some strong claims about providing deep theoretical explanation should be more moderate. In essence, it seems that the paper verifies *numerically* (in section B.3) that Figalli's theorem (stated in Appendix B) holds in this context.

2) This is just a suggestion. I think in some parts a lighter notation and a more intuitive explanation could help.

3) After Eq. (5) in the Appendix the authors mention Newton's method, and Thm 3 is also specific to Newton's method. Then they mention that *Gradient Descent* is used (and in the main part of the paper they mentioned Adam). This is confusing. All these algorithms are different, and Newton's method does not imply convergence results for gradient descent. I don't see how Thm 3 is relevant.

4) This is a simple doubt. To avoid non-differentiability of the gradient, the OT step computes the Brenier potential and is able to locate the singularities. I wonder if using a simpler approach through optimization for nosmooth problems (such as Moreau envelopes or proximal methods) could resolve this issue? In the negative case, why not?

5) Some Minor comments:
1. Define OT in the abstract (Optimal Transportation?)
2. What is AE? (not defined also; Auto Encoder?)
3. There are lots of typos through the text, such as missing "the", "a", etc.
and a couple mispelled words. I suggest the authors proofread the draft
more carefully.
4. pp. 4 ... what is a "PL convex function". PL is not defined.


**Experience Assessment:**

I have read many papers in this area.

**Review Assessment: Checking Correctness Of Derivations And Theory:**

I assessed the sensibility of the derivations and theory.

**Review Assessment: Checking Correctness Of Experiments:**

I assessed the sensibility of the experiments.

**Review Assessment: Thoroughness In Paper Reading:**

I read the paper at least twice and used my best judgement in assessing the paper.

---

> ### Author Response · Authors · 2019-11-09
> **Response to Reviewer #3**
>
>
>
> ----------------------------
> Q1: Although this paper brings a new perspective, based on optimal transport theory, as far as I can
> understand this paper does not establish formal new results. Thus I think some strong claims about
> providing deep theoretical explanation should be more moderate. In essence, it seems that the paper
> verifies *numerically* (in section B.3) that Figalli’s theorem (stated in Appendix B) holds in this
> context.
>
> Answer: This work focuses on using Figalli’s regularity theory of Optimal Transportation Map to
> explain mode collapse/mixture in generative models and propose a novel model to tackle it, not to
> develop the new regularity theorems. We will follow the reviewer’s suggestion to make our claims
> more moderate.
>
> ----------------------------
> Q2: This is just a suggestion. I think in some parts a lighter notation and a more intuitive explanation
> could help.
>
> Answer: We will follow reviewer’s suggestion to add more intuitive explanations and simplify the
> notations.
>
> ----------------------------
> Q3: After Eq. (5) in the Appendix the authors mention Newton’s method, and Thm 3 is also specific
> to Newton’s method. Then they mention that *Gradient Descent* is used (and in the main part of the
> paper they mentioned Adam). This is confusing. All these algorithms are different, and Newton’s
> method does not imply convergence results for gradient descent. I don’t see how Thm 3 is relevant.
>
> Answer: According to the variational framework of semi-discrete optimal transportation map,
> theorem 2, the computation of OT map is reduced to a convex optimization. Hence both gradient
> descend and Newton’s method converge. We will modify Thm 3 accordingly. Furthermore, this work
> focuses on gradient descend method, in the future work, we will explore Newton’s method as well.
>
> ----------------------------
> Q4: This is a simple doubt. To avoid non-differentiability of the gradient, the OT step computes the
> Brenier potential and is able to locate the singularities. I wonder if using a simpler approach through
> optimization for nosmooth problems (such as Moreau envelopes or proximal methods) could resolve
> this issue? In the negative case, why not?
>
> Answer: Although the OT map is discontinuous, and the Brenier potential is non-differentiable, the
> energy to be optimized is C^2 in terms of h. Therefore, in the current work, for the optimization
> purpose it is unnecessary to use Moreau envelope or proximal methods. Specifically, the convex
> energy E(h) we aim to optimize is differentiable with respect to h. With the optimal h, the OT map
> can be induced. Therefore for the optimization, we actually do not need smoothing techniques to carry
> out the optimization. Secondly, the non-differentiability of Brenier’s potiential uh(x) is considered
> with respect to x given the optimal h. This is independent of the optimization process.
>
> ----------------------------
> Q5: Some Minor comments: 1. Define OT in the abstract (Optimal Transportation?) 2. What is AE?
> (not defined also; Auto Encoder?) 3. There are lots of typos through the text, such as missing "the",
> "a", etc. and a couple mispelled words. I suggest the authors proofread the draft more carefully. 4.
> pp. 4 ... what is a "PL convex function". PL is not defined.
>
> Answer: We thank the reviewer 3 for the comments. In the paper, OT represents optimal transport,
> AE means autoencoder and PL is the abbreviation of piece-wise linear. We will add more explains to
> the abbreviations and find native speakers to help proofread the updated manuscript.

---

### Official Review · AnonReviewer4 · 2019-11-08
**Official Blind Review #4**

**Rating:** 3

**Review:**

This paper deals with an important problem of mode collapse and mode mixture. In order to
tackle the both problems, the paper proposes to separate the manifold embedding and the
optimal transportation problems; the first part being carried out using an autoencoder to map the
images onto the latent space and the second part is accomplished using a GPU-based
convex optimization to find the discontinuous transportation maps.

I have some doubts about moving from the "semi-discrete OT map" to the piece-wise linear extension. The illustration in Fig. 3, and implicit in all the explanation charts is the fact that discontinuity can be found by a linear separation. This seems to be an extremely simplifying assumption, which leads to not so great visual results from the paper. Although the numerical results seems promising, I feel that fewer images, but larger in size, and analysis of mode collapse phenomenon in real images would have been much better.

Singular set detection seems to be the most tricky part in this paper, which should have been explained further. The Simplex projection assumption, renders this part not that tricky, but that is where I feel the biggest doubt about this paper lies.

The authors themselves mention the need for a high quality auto encoder model to encode celebA dataset, which has been improved upon by numerous other papers, the claims seems not too strong. Also, the method does not have any adversarial training and hence, it studies the GAN idea from only fixing the generator point of view.

**Experience Assessment:**

I have published one or two papers in this area.

**Review Assessment: Checking Correctness Of Derivations And Theory:**

I assessed the sensibility of the derivations and theory.

**Review Assessment: Checking Correctness Of Experiments:**

I carefully checked the experiments.

**Review Assessment: Thoroughness In Paper Reading:**

I read the paper at least twice and used my best judgement in assessing the paper.

---

> ### Author Response · Authors · 2019-11-11
> **Response to Reviewer #4**
>
>
> ----------------------------
> Q1:  I have some doubts about moving from the "semi-discrete OT map" to the piece-wise linear extension.   The illustration in Fig. 3,  and implicit in all the explanation charts is the fact that discontinuity can be found by a linear separation. This seems to be an extremely simplifying assumption, which leads to not so great visual results from the paper.
>
> Answer: Here we want to clarify that singular set detection is *piece-wise linear* separation, rather than *linear* separation. In Fig. 3(a), the singular set (shown in red lines) is illustrated by a piece-wise linear curve. Also, Fig. 6 of the appendix shows another example with the numerically computed singular set (also piece-wise linear) by our method, and it is much more curved and complicated.
>
> ----------------------------
> Q2: Although the numerical results seems promising, I feel that fewer images, but larger in size, and analysis of mode collapse phenomenon in real images would have been much better.
>
> Answer:  As shown in the the last paragraph of Section 4.1, we conducted experiments of mode collapse on real images like stacked MNIST and CelebA on Section C.3 and C.4 of the appendix.
>
> ----------------------------
> Q3:  Singular set detection seems to be the most tricky part in this paper, the Simplex projection assumption, renders this part not that tricky, but that is where I feel the biggest doubt about this paper lies.
>
> Answer: (1) There is no simplex projection assumption in our paper. In fact, Fig.3(a) illustrates the Brenier potential and the corresponding power diagram. The upper hyperplane envelope in top of Fig.3(a) is the graph of Brenier potential, and the bottom of Fig. 3(a) shows the source domain of the Brenier potential, expressed as a cell decomposition structure. Each facet in the image of Brenier potential corresponds to a cell in the source domain (Ω), and the ridges on the image of Brenier potential corresponds to edges of cells in the source domain.
>
> (2) In the image of the Brenier potential, the "sharp ridges" are composed of the edges where the angles between the corresponding pairs of adjacent facets are large (as shown in Fig.3(a)). In fact, the normal of a facet n= (p_1, p_2,..., p_d, −1) actually corresponds to a latent code y= (p_1, p_2,..., p_d). And the large angle between two adjacent facets means that the distance between the corresponding latent codes is large.  This often happens when the codes come from different modes.  Thus, the singular set, or equivalently the "sharp ridges" gives the information about different modes.
>
> (3) Singular set detection is proposed in our paper for the following reason. Firstly, the singular set is totally decided by the semi-discrete OT map, or equivalently, the Brenier potential (Fig.  3(a)). Secondly, the image of the semi-discrete OT map itself is the given discrete latent code, thus we extend it with a piece-wise linear manner, so that the extended OT map can be used to *generate new codes* (Fig. 3(b)). Thirdly, the samples around the singular set will be mapped to the gaps among the modes by our extended OT map and cause the mode mixture problem, thus the singular detection is needed. Finally, given a sample x, if it falls around the singular set (checked by Alg. 2), we just don’t use it to generate new latent code.

---

> > ### Author Response · Authors · 2019-11-12
> > **Response to Reviewer #4**
> >
> >
> > ----------------------------
> > Q4: Singular set detection seems to be the most tricky part in this paper, which should have been explained further.
> >
> > Answer: In the following, we justify our algorithm using the theoretic works summarized in the following book:
> > Figalli, A. (2017). The Monge–Ampère equation and its applications.
> >
> > According to Brenier’s theorem, the optimal transportation map T is the gradient of the convex Brenier potential u, and u satisfies the Monge-Ampére equation.
> > In his book, the Fields medalist Figalli proved the existence and the uniqueness of the solution to the
> > Monge-Ampere equation in Chapter 2, where he used Alexandrov’s approach:
> > 1. Approximate the data distribution ν to a sequence Dirac distributions νn, such that the sequence of {ν_n} weakly converges to ν;
> > 2. For each Dirac measure ν_n, there exists an Alexandrov’s solution u_n, which is exactly the discrete Brenier potential in our paper;
> > 3. The weak solutions {u_n} converges to the real solution u, u is C^1 almost everywhere, except at the singular set.
> >
> > Our Semi-Discrete OT algorithm is completely equivalent to Alexandrov’s solution. In fact, the proof
> > in Figalli’s book is not constructive, (Alexandrov’s original proof is based on Algebraic topology), which doesn’t induce an computational algorithm. Therefore, the theorem 2 in the Appendix gives a variational framework to explicitly compute the discrete Brenier potential. By Figalli’s work, the discrete Brenier potential {u_n} converges to the smooth Brenier potential, which is C^1 except at the singular set. The piece-wise linear map in Fig.3(a) converges to the real optimal transportation map.
> >
> > The singular set is the non-differentiable points (only C^0 but not C^1) of the Brenier potentials, namely the ridges of the graph of u. This ridge structure becomes prominent and well-preserved in
> > the process of approximating u by piece-wise linear polyhedra {u_n} in Fig.3(a).
> >
> > Compared to Fig.3, Fig. 6 and Fig. 7 in the appendix gives better illustration for the singularity. The
> > original version of Fig. 6 is given by Figalli as the Fig. 3.2 in the following article,
> > Figalli, A. (2010). Regularity properties of optimal maps between nonconvex domains in the plane.
> > Communications in Partial Differential Equations, 35(3), 465-479.
> >
> > We can see that the singular set has complicated geometric and topological structures, which can not
> > be captured by linear separation, but still can be found by *piece-wise linear approximation*. In
> > fact, the optimal transport map shown in Fig. 6 is numerically computed by our algorithm, and the
> > singular set is piece-wise linear, approximating the singular set in the smooth case (shown as Fig. 3.2
> > in the above mentioned article).
> >
> > Next we show that the singular set structure of the smooth Brenier potential is well preserved by
> > our SDOT map. From chapter 2 in Figalli’s book, we know that the piece-wise linear functions un
> > (discrete Brenier potential) converges to the real smooth Brenier potential u, which is C^1
> > everywhere except at the singular points. Therefore the graph of the smooth Brenier potential has ridges, these ridge structure are well preserved during the piece-wise linear approximation by the discrete Brenier
> > potential.
> >
> > Therefore, singularity detection boils down to locate the ride structure of the graph of the discrete
> > Brenier potential, which is a convex polyhedron. The ridge on a convex polyhedron can be easily
> > found by computing the angles between each pair of adjacent facets (dihedral angles for 2D case).
> > Because the discrete Brenier potential is convex, its projection induces a power diagram, each cell is
> > convex. The dual of this power diagram gives the power Delaunay triangulation of training samples
> > (y_i’s) (Fig. 2 of the following article). This geometric interpretation of semi-discrete OT doesn’t
> > require the linear separation assumption. The relation among discrete Brenier potential, power
> > diagram and power Delaunay triangulation is explained in details in
> > Gu, X., Luo, F., Sun, J., & Yau, S. T. (2016). Variational principles for Minkowski type problems,
> > discrete optimal transport, and discrete Monge–Ampère equations. Asian Journal of Mathematics,
> > 20(2), 383-398.

---

> > > ### Author Response · Authors · 2019-11-12
> > > **Response to Reviewer #4**
> > >
> > >
> > > ----------------------------
> > > Q5: The authors themselves mention the need for a high quality auto encoder model to encode celebA dataset, which has been improved upon by numerous other papers, the claims seems not too strong. Also, the method does not have any adversarial training and hence, it studies the GAN idea from only fixing the generator point of view.
> > >
> > > Answer: The main goal of this work is to tackle mode collapse and mode mixture problems in general generative models, not only for GANs. Our work targets at analysis and improvement of generators in generic generative models, including VAEs and GANs. In fact, generators in these models tend to map a unimodal Gaussian to the complex data distribution, which will inevitably encounter the singularity problem proposed in our work. We thank the reviewer #4 for pointing out the ambiguity of our motivation. We have revised our abstract and introduction parts, which analyze the discontinuity problems encountered by GANs and VAEs. Then we propose a new generative model called AE-OT. Actually, in the original version of our paper, we have reviewed all the DNN based generative models in the related work part, and made comparisons with GANs, VAEs and other generated models in the experiment part.
> > >
> > > Because our main focus is to solve mode collapse/mixture problems, we didn’t apply the most
> > > advanced auto-encoder (AE). If the capacity of AE is insufficient, the result is not satisfying, such as
> > > the celebA dataset noticed by the reviewer. But, as we explained in section 4.2, the 3rd paragraph, if
> > > the capacity of AE is sufficient, our model outperform others.
> > >
> > > As recent GAN improvements mostly focus on the discriminator, our work complements these
> > > works by critically analyzing and making improvement on the generator. Future research on adding
> > > adversarial loss to our current model is also intriguing.

---

### Author Response · Authors · 2019-11-09
**Thanks for your careful comments**

We thank reviewers for carefully examine our work in such a short time.  Since our work involvesnon-trivial theories from optimal transportation, such as the brand new theorems of Figalli, andregularity theorems for Monge-Ampere equation, the review requires huge amount of efforts.  Wedeeply appreciate all reviewers from deep of our hearts. Since all the reviews and rebuttals will bepublic online, we prepared our rebuttal with great caution, and addressed all the questions raised byreviewers carefully

---

### Decision · Program_Chairs · 2019-12-19

**Decision:**

Accept (Poster)

**Comment:**

The authors present a different perspective on the mode collapse and mode mixture problems in GAN based on some recent theoretical results.

This is an interesting work. However, two reviewers have raised some concerns about the results and hence given a low rating of the paper. After reading the reviews and the rebuttal carefully I feel that the authors have addressed all the concerns of the reviewers. In particular, at least for one reviewer I felt that there was a slight misunderstanding on the reviewer's part which was clarified in the rebuttal. The concerns of R1 about a simpler baseline have also been addressed by the authors with the help of additional experiments. I am convinced that the original concerns of the reviewers are addressed. Hence, I recommend that this paper be accepted.

Having said that, I strongly recommend that in the final version, the authors should be a bit more clear in motivating the problem. In particular, please make it clear that you are only dealing with the generator and do not have an adversarial component in the training. Also, as suggested by R3 add more intuitive descriptions to make the paper accessible to a wider audience.